# Hessian-Free Natural Gradient Descent for Physics Informed Machine Learning

## Abstract

Physics-Informed Machine Learning (PIML) methods, such as Physics-Informed Neural Networks (PINNs), are notoriously difficult to optimize. Recent advances utilizing second-order optimization techniques, including natural gradient and Gauss-Newton methods, have significantly improved training accuracy over first-order methods. However, these approaches are computationally prohibitive, as they require evaluating, storing, and inverting large curvature matrices, limiting scalability to small networks. In this work, we propose a Hessian-Free Natural Gradient Descent frameworks that employs a matrix-free approximation of the Hessian-vector product. This approach circumvents the need for explicitly constructing the Hessian matrix and incorporates a novel preconditioning scheme that significantly enhances convergence rates. Our method enables scaling to large neural networks and complex PDEs with up to a millions parameters. Empirically, we demonstrate that our approach outperforms state-of-the-art optimizers, such as LBFGS and Adam, achieving orders-of-magnitude speedup and superior accuracy across various benchmark PDE problems.

## 1 Introduction

**Partial Differential Equations (PDEs)**   Partial Differential Equations (PDEs) are central to the mathematical modeling of complex physical systems, including fluid dynamics, thermodynamics, and material sciences. Traditional numerical methods, such as finite element and spectral methods, often require fine discretization of the physical domain to achieve high accuracy. These approaches can become computationally expensive, particularly in engineering applications where systems must be solved repeatedly under varying parameters or initial conditions. Recent advances in machine learning (ML) have shown promise in addressing these challenges by leveraging neural networks (NNs) as potential alternatives or enhancements to traditional numerical solvers Kovachki et al. (2021); Li et al. (2020).

**Physics-Informed Neural Networks (PINNs)**   PINNs are a machine learning tool to solve forward and inverse problems involving partial differential equations (PDEs) using a neural network ansatz. They have been proposed as early as Dissanayake & Phan-Thien (1994) and were later popularized by the works Raissi et al. (2019); Karniadakis et al. (2021). PINNs are a meshfree method designed for the seamless integration of data and physics. Applications include fluid dynamics Cai et al. (2021), solid mechanics Haghighat et al. (2021) and high-dimensional PDEs Hu et al. (2023) to name but a few areas of ongoing research. Despite their popularity, PINNs are notoriously difficult to optimize Wang et al. (2020) and fail to provide satisfactory accuracy when trained with first-order methods, even for simple problems Zeng et al. (2022); Müller & Zeinhofer (2023). Recently, second-order methods that use the function space geometry to design gradient preconditioners have shown remarkable promise in addressing the training difficulties of PINNs Zeng et al. (2022); Müller & Zeinhofer (2023); Ryck et al. (2024); Jnini et al. (2024); Müller & Zeinhofer (2024). However, these methods require solving a linear system in the network's high-dimensional parameter space at cubic computational iteration cost, which prohibits scaling such approaches and limits their applicability to small-scale neural-networks.

A promising solution to alleviate this computational burden is the use of matrix-free methods. These methods avoid the need for explicitly forming or inverting the Hessian matrix and instead approximate the curvature matrix implicitly via iterative procedures Schraudolph (2002). While matrix-free

approaches have been explored for PINNs Zeng et al. (2022); Bonfanti et al. (2024); Jnini et al. (2024); Zampini et al. (2024), their success is often limited in scale due to the ill-conditioning of the underlying optimization problems and the absence of straightforward scalable preconditioning scheme, resulting in prohibitively high iteration counts.

**Main Contributions**  Our main contributions can be summarized as follows:

- **Hessian-Free Natural Gradient Descent (HF-NGD):** We introduce a novel HF-NGD framework that leverages a matrix-free approximation of the Hessian. Our method incorporates a Matrix-Free low-rank preconditioning strategy based on the truncated eigenvalue decomposition of the Gauss-Newton Hessian to address the issue of ill-conditioning by reducing large gaps between eigenvalues. This significantly enhances the convergence rates of iterative solvers like Conjugate Gradient (CG).
- **Scalability to Large Networks and Complex PDEs:** Our approach scales second-order optimizers that respect the underlying function space geometry, enabling us to train neural networks with up to a million parameters and optimize neural operators. This results in state-of-the-art performance, achieving more than a 1-order-of-magnitude improvement compared to LBFGS across benchmarks and over a 2-orders-of-magnitude improvement on neural operators.

**Related Works**  Improving the training of PINNs has been the focus of extensive research. Early approaches explored adaptive re-weighting of loss terms for PDE residuals, data terms, and boundary conditions Wang et al. (2021), and adaptive sampling methods to optimize collocation points based on error indicators like the PDE residual Wu et al. (2023). While these techniques enhanced PINN training, they often failed to achieve relative $L^2$ errors below $10^{-4}$ for even simple problems.

Recent work has shown that second-order optimization methods, including LBFGS, significantly improve PINN accuracy. Notably, methods adopting an infinite-dimensional perspective have achieved near-single-precision accuracy for PINNs Zeng et al. (2022); Müller & Zeinhofer (2023); Ryck et al. (2024); Jnini et al. (2024); Zampini et al. (2024). However, their high per-iteration cubic cost—solving a large linear system in the network's parameter space—limits their scalability to small networks.

Matrix-free methods have been proposed to compute Gauss-Newton directions without explicitly forming the Hessian Martens (2010); Schraudolph (2002); Zeng et al. (2022); Jnini et al. (2024). Despite reducing computational costs, these methods suffer from ill-conditioning, leading to slow convergence for large networks without efficient preconditioners. Our algorithm addresses this by leveraging the structure of the curvature matrix to design a function space-based preconditioner, significantly improving inner solver convergence.

An alternative approach uses Kronecker-Factored Approximate Curvature (KFAC) Dangel et al. (2024) to scale second-order optimizers but requires problem-specific adaptations. Our method, in contrast, is both PDE- and architecture-agnostic.

Our idea of cutting large gaps within the leading eigenvalues of the Hessian spectrum is also aligned with recent advances in preconditioning techniques, such as volume sampling Rodomanov & Kropotov (2020), polynomial preconditioning Doikov & Rodomanov (2023), and spectral preconditioning Doikov et al. (2024). While these methods have shown promise for structured convex objectives, ours is the first to apply low-rank preconditioning in Gauss-Newton methods for PINN optimization.

## 2 PRELIMINARIES

### 2.1 PHYSICS-INFORMED NEURAL NETWORKS

For a given domain $\Omega \subset \mathbb{R}^d$ and a general PDE of the form

$$\mathcal{L}u = f \quad \text{in } \Omega, \quad u = g \quad \text{on } \partial\Omega,$$

where $\mathcal{L}$ is a differential operator, $f$ is the source term, and $g$ represents the boundary conditions, the solution $u$ is approximated using a neural network $u_\theta$, parameterized by $\theta$. The loss function is

defined as:

$$L(\theta) = \frac{1}{2N_\Omega} \sum_{n=1}^{N_\Omega} \left(\mathcal{L}u_\theta(x_n) - f(x_n)\right)^2 + \frac{1}{2N_{\partial\Omega}} \sum_{n=1}^{N_{\partial\Omega}} \left(u_\theta(x_n) - g(x_n)\right)^2, \tag{1}$$

where $\{x_n \in \Omega\}_{n=1}^{N_\Omega}$ are the collocation points in the interior of the domain and $\{x_n \in \partial\Omega\}_{n=1}^{N_{\partial\Omega}}$ are the boundary points.

First-order optimizers, such as gradient descent and Adam, often fail to provide satisfactory results when applied to Physics-Informed Neural Networks (PINNs) due to the ill-conditioning and non-convexity of the loss landscape, as well as the complexities introduced by the differential operator $\mathcal{L}$ Wang et al. (2020); Krishnapriyan et al. (2021). To overcome these challenges, we adopt the "optimize-then-discretize" paradigm, as advocated in Müller & Zeinhofer (2024). This approach formulates the optimization problem in the infinite-dimensional function space and only then discretizes it into the finite-dimensional parameter space of the neural network.

## 2.2 GAUSS-NEWTON NATURAL GRADIENT METHOD

Given its applicability to both linear and nonlinear PDEs, we focus on the Gauss-Newton Natural Gradient (GNNG) method for PINNs, introduced in Jnini et al. (2024) for the Navier-Stokes equation. When applied to PINN objectives like Eq. equation 1, GNNG corresponds to the Gauss-Newton method in parameter space.

GNNG mimics the Gauss-Newton method in function space up to a projection onto the model's tangent space and a discretization error that vanishes quadratically in the step size, thus providing locally optimal residual updates as shown in Jnini et al. (2024).

Natural gradient methods perform parameter updates via a preconditioned gradient descent scheme:

$$\theta \leftarrow \theta - \alpha \mathcal{H}(\theta)^+ \nabla L(\theta),$$

where $\mathcal{H}(\theta)^+$ denotes the pseudo-inverse of the Gauss-Newton Hessian $\mathcal{H}(\theta)$, and $\alpha$ is a step size.

In general, the Gauss-Newton Hessian $\mathcal{H}(\theta)$ for the PINN loss is formulated as:

$$\mathcal{H}(\theta) = \frac{1}{N} \sum_{n=1}^{N} DR(u_\theta)[\partial_{\theta_i} u_\theta](x_n) DR(u_\theta)[\partial_{\theta_j} u_\theta](x_n),$$

where $DR(u_\theta)$ is the Fréchet derivative of the residual operator $R$ which corresponds to both the PDE residual and the boundary conditions. When applied to PINN objectives like Eq. equation 1, the residuals are defined as follows: the interior residual is given by $r_{\Omega,n}(\theta) = \mathcal{L}u_\theta(x_n) - f(x_n)$, for $x_n \in \Omega$, and the boundary residual is $r_{\partial\Omega,n}(\theta) = u_\theta(x_n) - g(x_n)$, for $x_n \in \partial\Omega$. This gives us the full Gauss-Newton Hessian, which consists of two components:

$$\mathcal{H}(\theta) = \mathcal{H}_\Omega(\theta) + \mathcal{H}_{\partial\Omega}(\theta),$$

## 2.3 MATRIX-FREE COMPUTATION

The correspondence between the Gauss-Newton method in function space and parameter space is shown in Müller & Zeinhofer (2024), where it is demonstrated that the following relation holds:

$$\mathcal{H}(\theta) = J(\theta)^\top J(\theta),$$

where $J(\theta)$ denotes the Jacobian of the corresponding residual $r(\theta)$. This equivalence ensures that applying Gauss-Newton's method to the discretized residual agrees with the function space algorithm, provided the same quadrature points are used in the discretization of $\mathcal{H}$. This formulation allows us to compute the action of the Gauss-Newton Hessian $\mathcal{H}(\theta)$ on a vector $v$ in a matrix-free manner, using a combination of forward and backward mode automatic differentiation, which requires only a constant overhead compared to a gradient computation Schraudolph (2002). Specifically, we compute $\mathcal{H}(\theta)v = J^\top w$, where $w = Jv$.

With these Hessian-vector products, we can employ matrix-free solvers like the conjugate gradient method Trefethen & Bau (2022) to efficiently compute $\mathcal{H}(\theta)^\dagger \nabla L(\theta)$. This approach circumvents

the prohibitive cubic computational cost of direct methods, enabling scalable optimization in high-dimensional parameter spaces.

However, as described in Jnini et al. (2024) and demonstrated experimentally, the naive matrix-free approach suffers from severe ill-conditioning. Without appropriate preconditioning, conjugate gradient solvers require a prohibitively high number of iterations to converge. We address this issue by proposing a matrix-free preconditioning strategy that takes advantage of the particular structure of the Gauss-Newton Hessian, and that scales to networks with millions of parameters.

## 3 MATRIX-FREE PRECONDITIONING OF THE GAUSS-NEWTON HESSIAN

The efficiency of the Conjugate Gradient (CG) method in solving linear systems depends on the spectral properties of the matrix. The number of iterations $k$ required to reduce the initial error $\|e_0\|_A$ by a factor $\varepsilon$ when solving $Ax = b$ using CG satisfies Saad (2003):

$$k \leq \frac{1}{2}\sqrt{\kappa(A)} \log\left(\frac{\|e_0\|_A}{\varepsilon}\right),$$

where $\kappa(A) = \frac{\lambda_{\max}(A)}{\lambda_{\min}(A)}$ is the condition number of $A$ in the $A$-norm. For ill-conditioned matrices, $\kappa(A)$ is large, resulting in slow convergence of CG. In our case, the Gauss-Newton Hessian $\mathcal{H}$, derived from the discretized PDE residuals, is typically ill-conditioned due to the wide range of scales in the eigenvalues.

### 3.1 SPECTRAL ANALYSIS OF THE GAUSS-NEWTON GAUSS-NEWTON HESSIAN

In practice, the eigenvalue distribution of the Gauss-Newton Hessian frequently exhibits a small number of significantly large eigenvalues, followed by a rapid decay to much smaller values. This indicates that only a few directions in the parameter space are associated with high curvature, while the majority of directions have low curvature. This spectral structure can substantially affect the efficiency of CG methods, as they tend to converge slowly in directions corresponding to small eigenvalues.

To further illustrate the spectral properties of the Gauss-Newton Hessian, we analyze the Kovasznay flow, from the experiment described in section4.1. For this problem, the Gauss-Newton Hessian matrix $\mathcal{H}$ is observed to be rank-deficient, with only a small number of large eigenvalues dominating the spectrum. This indicates that the problem is effectively low-dimensional, with only a few directions in the parameter space contributing significantly to the residual minimization.

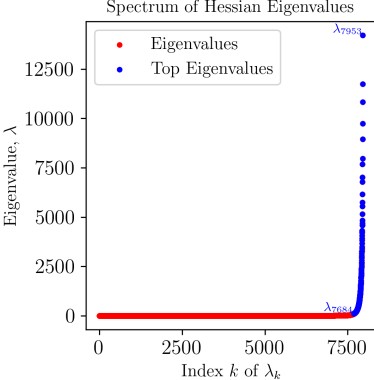

Figure 1: Spectrum of the Gauss-Newton Hessian for a multi-layer perceptron (MLP) with parameter size $n = 7953$. The red dots represent all the eigenvalues, while the highlighted blue dots correspond to the eigenvalues larger than 100. The indices $\lambda_{7684}$ and $\lambda_{7953}$ indicate the first and last of these top eigenvalues, respectively.

## 3.2 PRECONDITIONING STRATEGY

The observed spectral properties of the Gauss-Newton Hessian suggest that the matrix possesses a few large eigenvalues followed by a rapidly decaying tail of smaller eigenvalues (see Figure 1). This indicates that the ill-conditioning is primarily due to a small number of dominant eigenvalues, while the majority of the spectrum corresponds to directions with smaller curvature.

To address the ill-conditioning and improve the convergence of the Conjugate Gradient (CG) solver, we propose a preconditioning strategy that effectively *truncates* the spectrum by approximating $\mathcal{G}$ with a low-rank matrix capturing the dominant eigenvalues. We approximate the Gauss-Newton Hessian $\mathcal{G}$ using its truncated eigenvalue decomposition with a damping factor:

$$M_k = V_k \Lambda_k V_k^\top + \lambda I, \quad \text{where} \quad V_k \in \mathbb{R}^{n \times k}, \ \Lambda_k \in \mathbb{R}^{k \times k},$$

and $\lambda$ is a damping parameter.

To efficiently compute the action of the inverse preconditioner $M_k^{-1}$ on a vector without explicitly forming $M_k$, we employ the Sherman-Morrison-Woodbury formula:

$$M_k^{-1} = \frac{1}{\lambda} I - \frac{1}{\lambda^2} V_k D^{-1} V_k^\top, \quad D = \left( \Lambda_k^{-1} + \frac{1}{\lambda} I_k \right)^{-1},$$

where $I_k$ is the $k \times k$ identity matrix. To compute $M_k^{-1} v$ for any vector $v$, we use:

$$M_k^{-1} v = \frac{1}{\lambda} v - \frac{1}{\lambda^2} V_k D V_k^\top v.$$

The steps for applying $M_k^{-1}$ are as follows:

- Compute $z = V_k^\top v$ (requiring $k$ dot products),
- Compute $y = Dz$ (element-wise multiplication since $D$ is diagonal),
- Compute $w = V_k y$,
- Combine results: $M_k^{-1} v = \frac{1}{\lambda} v - \frac{1}{\lambda^2} w$.

Since we only store $V_k$ and $\Lambda_k$ (both of size $n \times k$ and $k \times k$, respectively), and $k \ll n$, the memory requirements are minimaland reduce to $\mathcal{O}(kn)$ memory complexity.

---

**Condition Number of the Preconditioned System**

Let $\mathcal{G} \in \mathbb{R}^{n \times n}$ be a symmetric positive definite matrix with eigenvalues $\tilde{\lambda}_1 \geq \tilde{\lambda}_2 \geq \cdots \geq \tilde{\lambda}_n > 0$ and corresponding orthonormal eigenvectors $\{\mathbf{v}_i\}_{i=1}^n$, such that:

$$\mathcal{G} = V \tilde{\Lambda} V^\top, \quad V = [\mathbf{v}_1, \mathbf{v}_2, \ldots, \mathbf{v}_n], \quad \tilde{\Lambda} = \text{diag}(\tilde{\lambda}_1, \tilde{\lambda}_2, \ldots, \tilde{\lambda}_n).$$

Consider the preconditioner $M_k$ defined by:

$$M_k = V_k \Lambda_k V_k^\top + \lambda I,$$

where:

- $V_k = [\mathbf{v}_1, \mathbf{v}_2, \ldots, \mathbf{v}_k] \in \mathbb{R}^{n \times k}$,
- $\Lambda_k = \text{diag}(\tilde{\lambda}_1, \tilde{\lambda}_2, \ldots, \tilde{\lambda}_k) \in \mathbb{R}^{k \times k}$,
- $\lambda > 0$ is a damping parameter.

Then, the exact condition number $\kappa(M_k^{-1} \mathcal{G})$ of the preconditioned system $M_k^{-1} \mathcal{G}$ is:

$$\kappa(M_k^{-1} \mathcal{G}) = \frac{\tilde{\lambda}_{k+1}(\tilde{\lambda}_k + \lambda)}{\tilde{\lambda}_k \lambda}.$$

---

In the working limit where $\tilde{\lambda}_k \gg \lambda$, the condition number of the preconditioned system simplifies to $\kappa(M_k^{-1} \mathcal{G}) \approx \frac{\tilde{\lambda}_{k+1}}{\lambda}$. This implies that the preconditioner effectively cuts the top $k$ eigenvalues, leaving the condition number dependent on the ratio $\frac{\tilde{\lambda}_{k+1}}{\lambda}$. We can thus design an online control Algorithm for the condition number of the Hessian.

## 3.3 ONLINE CONDITION NUMBER CONTROL ALGORITHM

We propose an algorithm that controls the condition number of the Gauss-Newton Hessian $\mathcal{H}$ online by dynamically adjusting the rank of its low-rank decomposition.

To efficiently approximate the low-rank structure of the Gauss-Newton Hessian $\mathcal{H}_k$, we employ a hybrid approach combining Lanczos decomposition with orthogonal iteration. Lanczos captures dominant eigenvalues, while orthogonal iteration refines the subspace between Lanczos updates. For details of the Lanczos and Orthogonal Iteration algorithms, refer to Appendix 3 and Appendix 4, respectively.

**Lanczos Decomposition:** Lanczos is performed periodically (every $N_L$ iterations) to generate a Krylov subspace that approximates the largest eigenvalues and eigenvectors of $\mathcal{H}_k$. Due to its optimal convergence behavior, especially for the largest eigenvalues, Lanczos often exhibits superlinear convergence Saad (2003).

**Orthogonal Iteration:** Orthogonal iteration refines the subspace $\hat{V}_{k-1}$ between Lanczos updates, adapting to changes in $\mathcal{H}_k$ as parameters are updated. Its linear convergence is offset by its low computational cost, making it suitable for maintaining subspace accuracy without incurring the higher cost of a full Lanczos decomposition at each step.

**Adaptive Low-Rank Decomposition and condition number control:** Our algorithm adaptively alternates between full Lanczos decomposition and orthogonal iteration, depending on the epoch and the residual norm. The rank of the subspace is dynamically adjusted by monitoring the smallest eigenvalue of the previous Ritz values. When this eigenvalue surpasses a threshold tied to the damping factor—which serves as an approximation of the condition number—the rank is incremented.

---

**Algorithm 1** Adaptive Low-Rank Decomposition with Condition Number Control

---

**Require:** Current epoch $e$, previous subspace $\hat{V}_{k-1}$, previous Ritz values $\Lambda_{k-1}$, Lanczos update interval $N_L$, tolerance $\epsilon$, maximum orthogonal iterations $T$, initial rank $k$, damping factor $\lambda$, max admissible condition number $\alpha$, rank increment $\Delta k$, maximum rank $k_{\max}$

**Ensure:** Updated subspace $\hat{V}_k$, updated Ritz values $\Lambda_k$, updated rank $k$

1: **if** $e \bmod N_L = 0$ **then**
2:     **Check the smallest previous Ritz eigenvalue:**
3:     $\lambda_{\min} = \min(\Lambda_{k-1})$ {Get the smallest eigenvalue from the previous iteration}
4:     **if** $\lambda_{\min} > \alpha \cdot \lambda$ **then**
5:         Increase rank $k \leftarrow \min(k + \Delta k, k_{\max})$ {Increase rank by $\Delta k$ if smallest previous eigenvalue exceeds threshold}
6:     **end if**
7:     Perform Lanczos decomposition to obtain $\hat{V}_k$, $\Lambda_k$ {Ritz values for the current epoch}
8: **else**
9:     Compute residual norm $r = \left\| \mathcal{H}_k \hat{V}_{k-1} - \hat{V}_{k-1}\Lambda_{k-1} \right\|_F$
10:     **if** $r > \epsilon$ **then**
11:         Perform Lanczos decomposition to obtain $\hat{V}_k$, $\Lambda_k$
12:     **else**
13:         Update the subspace $\hat{V}_k$ using Orthogonal Iteration
14:         Compute updated Ritz values $\Lambda_k$ based on the new subspace $\hat{V}_k$ {In practice, old Ritz values provide a satisfying approximation}
15:     **end if**
16: **end if**

---

## 3.4 OPTIMIZATION WORKFLOW: HESSIAN-FREE NATURAL GRADIENT WITH LINE SEARCH AND LOW-RANK PRECONDITIONING

Given a partial differential equation (PDE) defined by a differential operator $\mathcal{L}$ and boundary conditions , with a corresponding neural network ansatz for $u_\theta$, we employ the following optimization procedure. The objective is to minimize the loss function, as described in Eq. equation 1, using

Hessian-Free Natural Gradient optimization with low-rank preconditioning and line search. The workflow is detailed below in Algorithm 2:

---

**Algorithm 2** Hessian-Free Natural Gradient with Line Search and Low-Rank Preconditioning

---

1: **Input:** initial parameters $\theta_0 \in \Theta$, maximum number of iterations $N_{\max}$, Lanczos update interval $N_L$, tolerance $\epsilon$, maximum orthogonal iterations $T$
2: Initialize subspace $\hat{V}_0$ and Ritz values $\Lambda_0$
3: **for** $k = 1$ to $N_{\max}$ **do**
4:     Compute gradient $\nabla L(\theta_{k-1})$, where $L(\theta)$ is the loss function in Eq. equation 1
5:     Perform Adaptive Low-Rank Decomposition to obtain $\hat{V}_k$, $\Lambda_k$ {Refer to Algorithm 1}
6:     Construct the low-rank approximation $H_k = \hat{V}_k \Lambda_k \hat{V}_k^\top$ {Matrix-free Hessian approximation}
7:     Define the preconditioning operator $M_k^{-1}$ via the Woodbury matrix identity:

$$M_k^{-1} = \frac{1}{\alpha_k} I_n - \frac{1}{\alpha_k^2} \hat{V}_k \left( \Lambda_k^{-1} + \frac{1}{\alpha_k} I_k \right)^{-1} \hat{V}_k^\top$$

8:     Use the Preconditioned Conjugate Gradient (PCG) method with $M_k^{-1}$ as the preconditioner to solve:
$$(H(\theta_{k-1}) + \alpha_k I)s_k = \nabla L(\theta_{k-1})$$

9:     Perform line search to determine step size $\eta_k$:
$$\eta_k = \arg \min_{\eta \in [0,1]} L(\theta_{k-1} - \eta s_k)$$

10:     Update parameters $\theta_k = \theta_{k-1} - \eta_k s_k$
11: **end for**

---

## 4 EXPERIMENTS

We evaluate the performance and versatility of our proposed **Hessian-Free Gauss-Newton Natural Gradient** (HF-NGD) optimization algorithm across diverse benchmark PDEs. These experiments are designed to demonstrate the algorithm's robustness to varying PDE complexities and neural network architectures. Table 4 summarizes the residuals and their linearized forms, and detailed hyperparameter settings and additional implementation specifics are provided in Appendix B.

We report the results of our experiments using the runs with the lowest relative $L^2$ error across 10 different initialization seeds. A statistical breakdown of all runs is available in Appendix B. All experiments were conducted on a compute cluster equipped with NVIDIA A100 GPUs (80GiB RAM) in double precision, using the JAX library Bradbury et al. (2018). Each optimizer was given the same computation time budget.

We summarize the results in Table 1, and an additional experiment with the Allen-Cahn equation is provided in Appendix B.

Table 1: Best relative $L^2$ error across different solvers for benchmark experiments within the allocated time budget. Best-performing solver is highlighted.

| Experiment | HF-NGD | ADAM | LBFGS | SGD |
|---|---|---|---|---|
| Allen-Cahn | $\mathbf{5.5788 \times 10^{-4}}$ | $9.1279 \times 10^{-3}$ | $9.1654 \times 10^{-4}$ | $5.7104 \times 10^{-1}$ |
| Klein-Gordon | $\mathbf{3.8747 \times 10^{-5}}$ | $2.0676 \times 10^{-3}$ | $1.1752 \times 10^{-4}$ | $1.0860 \times 10^{-3}$ |
| Poisson (10D) | $\mathbf{3.2020 \times 10^{-5}}$ | $4.2226 \times 10^{-3}$ | $3.2373 \times 10^{-4}$ | $5.0698 \times 10^{-3}$ |
| Navier-Stokes | $\mathbf{9.0018 \times 10^{-5}}$ | $9.0281 \times 10^{-3}$ | $1.8595 \times 10^{-3}$ | $1.8706 \times 10^{-2}$ |

### 4.1 EFFICACY OF THE PRECONDITIONING: 2D NAVIER-STOKES

We demonstrate the efficacy of our proposed preconditioning strategy using the two-dimensional steady Navier-Stokes flow described by Kovasznay (1948), with Reynolds number $Re = 40$, over $\Omega = [-0.5, 1.0] \times [-0.5, 1.5]$. The analytical solutions for the velocity and pressure fields are:

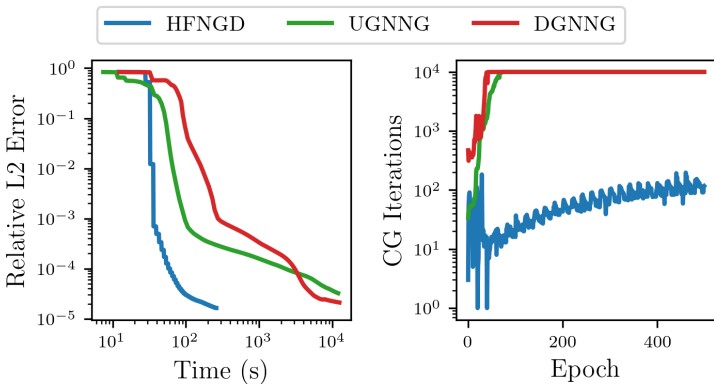

Figure 2: Relative $L^2$ Error vs. Time and CG Iterations vs. Epoch for different methods on the Kovasznay flow problem. The proposed preconditioning approach shows significantly faster convergence.

$$u^*(x,y) = 1 - e^{\lambda x}\cos(2\pi y), \quad v^*(x,y) = \frac{\lambda}{2\pi}e^{\lambda x}\sin(2\pi y), \quad p^*(x,y) = \frac{1}{2}\left(1 - e^{2\lambda x}\right),$$

where $\lambda = \frac{1}{2} - \sqrt{\frac{1}{4} + 4\pi^2}$ and $\nu = \frac{1}{Re}$.

The neural network model used is a Multi-Layer Perceptron (MLP) with 4 hidden layers, each containing 100 neurons, resulting in a total of 30,903 trainable parameters. The neural network model used is a Multi-Layer Perceptron (MLP) with 4 hidden layers, each containing 100 neurons, resulting in a total of 30,903 trainable parameters.

We evaluate the performance of our proposed preconditioning strategy in comparison with two alternative methods. The first method is UGNNG, which is the unpreconditioned Gauss-Newton natural gradient descent using Conjugate Gradient (CG), where no preconditioning is applied. The second method is DGNNG, which utilizes diagonal preconditioning. This method approximates the Gauss-Newton Hessian by computing a diagonal preconditioner to accelerate CG convergence. The diagonal is calculated as:

$$\text{diag}(\mathcal{H}) = \frac{1}{|x|}\sum_{i=1}^{|x|}\sum_{j=1}^{o}\left(\frac{\partial \text{res}(\theta, x_i)}{\partial \theta}^{\top} e_j\right)^2,$$

where $|x|$ is the batch size, $o$ is the output dimension, and $e_j$ is the $j$-th unit vector.

Figure 2 compares the performance of each method in terms of relative $L^2$ error versus time , as well as CG iterations per epoch over a maximum of 10,000 iterations. Our proposed preconditioning strategy (HF-NGD) achieves significantly faster convergence compared to both (UGNNG) and DGNNG proposed in Martens (2010). This performance improvement is primarily attributed to the fact that the Gauss-Newton Hessian matrix is not diagonally dominant, making the diagonal preconditioning less effective. As a result, our method demonstrates more robust convergence behavior, especially for larger models or more complex PDEs.

### 4.2 KLEIN-GORDON EQUATION

The Klein-Gordon equation is a non-linear hyperbolic PDE, commonly used in applied physics to model relativistic wave propagation. The inhomogeneous Klein-Gordon equation is given by:

$$\frac{\partial^2 u}{\partial t^2} - \Delta u + u^2 = f, \quad x \in \Omega, \, t \in \Gamma,$$

where $\Omega = [-5, 5]^2$ is the spatial domain and $\Gamma = [0, 1]$ is the temporal domain. The initial conditions are:

$$u(x, 0) = \cos(\pi x_1)\cos(\pi x_2), \quad \frac{\partial u}{\partial t}(x, 0) = 0.$$

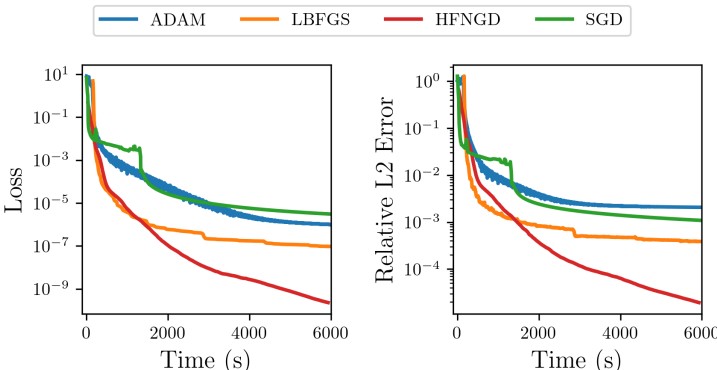

Figure 3: Loss and Relative $L^2$ Error plotted against time for different methods on the Klein-Gordon equation for the best run from each optimizer. Our method achieves significantly faster convergence compared to all other optimizers.

For error measurement, we use a manufactured solution:

$$u(x, t) = \cos(\pi x_1) \cos(\pi x_2) \cos(t).$$

The boundary conditions $u_{bc}(x)$ are derived from this exact solution.

The network used is a separable Physics-Informed Neural Network (PINN) architecture, as proposed in Cho et al. (2023), with 4 hidden layers and 128 neurons per layer for each spatial dimension, resulting in approximately 149,376 trainable parameters.

Figure 3 shows the superior convergence of HF-NGD compared to other optimizers, achieving up to an order of magnitude improvement in accuracy compared to LBFGS.

### 4.3 POISSON EQUATION IN 10 DIMENSIONS

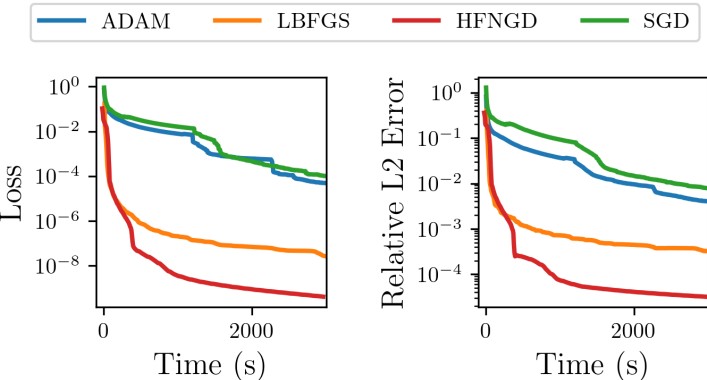

Figure 4: Loss and Relative L2 Error plotted against Time for different methods on the Poisson equation. HF-GNNG achieves the fastest convergence out of all optimizers.

We consider a 10-dimensional Poisson equation, defined as $-\Delta u = f(x)$, $x \in [0, 1]^{10}$, where $f(x)$ is the source term derived from the analytical solution $u^*(x) = \sum_{k=1}^{5} x_{2k-1} \cdot x_{2k}$. This analytical solution is used as a reference for training.

We utilize a standard MLP with 5 hidden layers, each containing 512 neurons and Tanh activations, resulting in approximately 1,102,849 trainable parameters. In this experiment, as shown in Figure 6 HF-GNNG achieves up to one order of magnitude improvement in performance compared to other optimizers.

## 4.4 Optimization of a Neural Operator for the Vorticity Equation

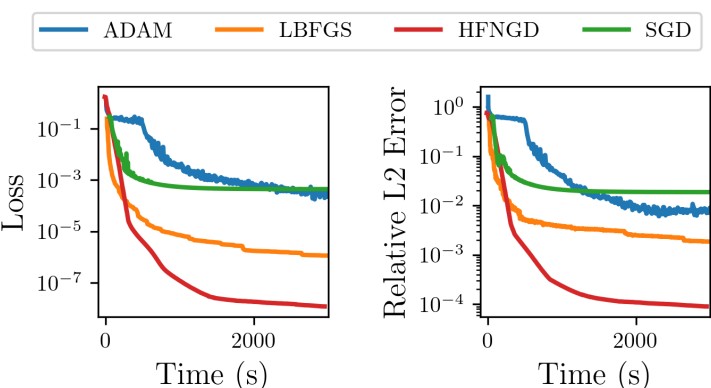

Figure 5: Loss and Relative L2 Error plotted against Time for different methods on the vorticity equation with a DeepONet for the best run out of each optimizer. Our optimizer noticeably achieves the fastest convergence out of all optimizers.

We apply our optimization algorithm to a neural operator framework trained with a purely physics-informed loss for the vorticity formulation of the Navier-Stokes equation in the Kovasznay flow configuration, as described in Section 4.1. The steady vorticity equation is:

$$u \cdot \nabla \omega - \nu \Delta \omega = 0,$$

where $\omega$ represents the vorticity, $u$ is the velocity field, and $\nu$ is the kinematic viscosity.

We train a DeepONet, consisting of two subnetworks: a branch network and a trunk network. The branch network, which encodes the Reynolds number, has 6 hidden layers with 100 neurons each. The trunk network, which encodes the spatial coordinates with periodic embeddings, also has 6 hidden layers with 100 neurons each. This configuration results in a total of approximately 101,500 trainable parameters. The network is trained without labeled data, relying entirely on a physics-informed loss that enforces the residuals of the governing PDE across a range of Reynolds numbers from 50 to 250. After training, the network's generalization capability is evaluated by inferring the solution for an unseen Reynolds number of $Re = 500$.

We observe that formulating the vorticity problem as a function-to-function mapping, using the DeepONet architecture, leads to a smoother loss landscape. This enhances the effectiveness of our HF-NGD method, allowing it to converge more rapidly compared to other optimizers. Our approach demonstrates up to two orders of magnitude improvement in accuracy compared to LBFGS within the time budget, as shown in Figure 5. This result highlights the potential of our method for solving parametric PDEs with high accuracy.

## Conclusion and Limitations

We introduced a Hessian-Free Natural Gradient Descent (HF-NGD) framework for Physics-Informed Machine Learning, leveraging a matrix-free Gauss-Newton Hessian approximation to significantly reduce computational overhead. Our low-rank preconditioning scheme based on truncated eigenvalue decomposition mitigates Hessian ill-conditioning and accelerates convergence. Empirical results on diverse PDE benchmarks—including Allen-Cahn, Klein-Gordon, Navier-Stokes, and high-dimensional Poisson equations—demonstrate that HF-NGD outperforms optimizers such as Adam and LBFGS, providing faster convergence with superior accuracy, achieving up to a 2-order-of-magnitude accuracy improvement in benchmark performance.

While our approach scales to large neural networks with up to a million parameters and neural operators, it is still limited by the memory complexity of $\mathcal{O}(kn)$, where $k$ is the rank and $n$ is the number of parameters. Scaling to evel larger networks may require exploring stochastic methods to estimate spectra or other efficient approximation techniques, which we consider as future work.

## REPRODUCIBILITY STATEMENT

All efforts have been made to ensure the reproducibility of the results presented in this paper. The theoretical contributions, including algorithm derivations and proofs, are provided in the appendix for clarity and completeness. We have also included all hyperparameters and network configurations used in the experiments to facilitate replication of the results.

Statistical data, such as the best, worst, and median performance metrics, are provided for each experiment across multiple optimization algorithms. In addition, we provide a working example of the proposed algorithm, included as supplementary material, to ensure the reproducibility of the results in a standard machine learning environment.

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

## A APPENDIX

### A.1 LANCZOS ALGORITHM

---
**Algorithm 3** Lanczos Algorithm

---
**Require:** Symmetric matrix $A \in \mathbb{R}^{n \times n}$, initial vector $v_0 \in \mathbb{R}^n$ with $\|v_0\| = 1$, maximum number of iterations $k$
**Ensure:** Approximation of $A$'s largest $k$ eigenvalues and corresponding eigenvectors
1: $v_1 \leftarrow 0, \beta_0 \leftarrow 0$
2: $V_1 \leftarrow [v_0]$ {Initialize orthonormal basis for Krylov subspace}
3: **for** j = 1 to k **do**
4:     $w_j \leftarrow Av_{j-1} - \beta_{j-1}v_{j-2}$
5:     $\alpha_j \leftarrow v_{j-1}^\top w_j$
6:     $w_j \leftarrow w_j - \alpha_j v_{j-1}$ {Orthogonalize against previous vectors}
7:     $\beta_j \leftarrow \|w_j\|$
8:     **if** $\beta_j = 0$ **then**
9:        **break** {If $w_j$ is zero, terminate early}
10:     **end if**
11:     $v_j \leftarrow w_j/\beta_j$
12:     $V_{j+1} \leftarrow [V_j, v_j]$ {Expand Krylov subspace}
13: **end for**
14: Form tridiagonal matrix $T_k$ with $\alpha_j$'s on the diagonal and $\beta_j$'s on the off-diagonals.
15: Compute eigenvalues and eigenvectors of $T_k$ to approximate the leading eigenvalues of $A$.
16: **return** Eigenvalues $\lambda_1, \ldots, \lambda_k$ and corresponding eigenvectors.

---

### A.2 ORTHOGONAL ITERATION

---
**Algorithm 4** Orthogonal Iteration

---
**Require:** Symmetric matrix $A \in \mathbb{R}^{n \times n}$, initial orthonormal matrix $V \in \mathbb{R}^{n \times k}$, tolerance $\epsilon_M$, maximum iterations $T$
**Ensure:** Approximate leading $k$ eigenvectors of $A$
1: QR-factorize $VR = Z$ for the starting matrix $Z$
2: **for** $k = 1$ to $T$ **do**
3:     $Y \leftarrow AV$
4:     $H \leftarrow V^T Y$
5:     **if** $\|Y - VH\|_2 \le \epsilon_M$ **then**
6:        **stop**
7:     **end if**
8:     QR-factorize $VR = Y$
9: **end for**

---

### A.3 PROOF OF LEMMA 3.2

*Proof.* We aim to compute the eigenvalues of $M_k^{-1}\mathcal{G}$ and determine the exact condition number $\kappa(M_k^{-1}\mathcal{G}) = \frac{\lambda_{\max}}{\lambda_{\min}}$.

**Step 1: Eigenvalue Decomposition of $\mathcal{G}$**

Since $\mathcal{G}$ is symmetric positive definite, it admits the eigenvalue decomposition:

$$\mathcal{G} = V\tilde{\Lambda}V^\top = \sum_{i=1}^{n} \tilde{\lambda}_i \mathbf{v}_i \mathbf{v}_i^\top,$$

where $V$ is an orthogonal matrix whose columns are the eigenvectors $\mathbf{v}_i$, and $\tilde{\Lambda}$ is a diagonal matrix containing the eigenvalues $\tilde{\lambda}_i$.

**Step 2: Construction of the Preconditioner $M_k$**

The preconditioner $M_k$ is constructed using the top $k$ eigenvectors and eigenvalues of $\mathcal{G}$, along with the damping term $\lambda I$:

$$M_k = V_k \Lambda_k V_k^\top + \lambda I = \sum_{i=1}^{k} \tilde{\lambda}_i \mathbf{v}_i \mathbf{v}_i^\top + \lambda \sum_{i=1}^{n} \mathbf{v}_i \mathbf{v}_i^\top - \lambda \sum_{i=1}^{k} \mathbf{v}_i \mathbf{v}_i^\top = \sum_{i=1}^{k} (\tilde{\lambda}_i - \lambda) \mathbf{v}_i \mathbf{v}_i^\top + \lambda I.$$

Simplifying:

$$M_k = \sum_{i=1}^{k} (\tilde{\lambda}_i + \lambda) \mathbf{v}_i \mathbf{v}_i^\top + \lambda \sum_{i=k+1}^{n} \mathbf{v}_i \mathbf{v}_i^\top.$$

**Step 3: Eigenvalues of $M_k$**

From the above expression, the eigenvalues $m_i$ of $M_k$ are:

$$m_i = \begin{cases} \tilde{\lambda}_i + \lambda, & \text{for } i = 1, \dots, k, \\ \lambda, & \text{for } i = k+1, \dots, n. \end{cases}$$

**Step 4: Eigenvalues of $M_k^{-1}\mathcal{G}$**

Since $\mathcal{G}$ and $M_k$ share the same eigenvectors $\mathbf{v}_i$, the action of $M_k^{-1}\mathcal{G}$ on $\mathbf{v}_i$ is:

$$M_k^{-1}\mathcal{G}\mathbf{v}_i = M_k^{-1}(\tilde{\lambda}_i \mathbf{v}_i) = \tilde{\lambda}_i M_k^{-1}\mathbf{v}_i.$$

But $M_k \mathbf{v}_i = m_i \mathbf{v}_i$, so:

$$M_k^{-1}\mathbf{v}_i = \frac{1}{m_i}\mathbf{v}_i.$$

Therefore, the eigenvalues $\mu_i$ of $M_k^{-1}\mathcal{G}$ are:

$$\mu_i = \frac{\tilde{\lambda}_i}{m_i}.$$

Substituting $m_i$ from Step 3:

$$\mu_i = \begin{cases} \dfrac{\tilde{\lambda}_i}{\tilde{\lambda}_i + \lambda}, & \text{for } i = 1, \dots, k, \\[2ex] \dfrac{\tilde{\lambda}_i}{\lambda}, & \text{for } i = k+1, \dots, n. \end{cases}$$

**Step 5: Determining the Maximum and Minimum Eigenvalues**

**Maximum Eigenvalue $\lambda_{\max}$:**

- For $i = k+1, \dots, n$:

$$\mu_i = \frac{\tilde{\lambda}_i}{\lambda} \leq \frac{\tilde{\lambda}_{k+1}}{\lambda},$$

since $\tilde{\lambda}_i \leq \tilde{\lambda}_{k+1}$. - For $i = 1, \dots, k$:

$$\mu_i = \frac{\tilde{\lambda}_i}{\tilde{\lambda}_i + \lambda} < 1.$$

Therefore, the maximum eigenvalue is:

$$\lambda_{\max} = \frac{\tilde{\lambda}_{k+1}}{\lambda}.$$

**Minimum Eigenvalue $\lambda_{\min}$:**

- For $i = 1, \ldots, k$:

$$\mu_i = \frac{\tilde{\lambda}_i}{\tilde{\lambda}_i + \lambda} \geq \frac{\tilde{\lambda}_k}{\tilde{\lambda}_k + \lambda},$$

since $\tilde{\lambda}_i \geq \tilde{\lambda}_k$. - For $i = k + 1, \ldots, n$:

$$\mu_i = \frac{\tilde{\lambda}_i}{\lambda} \geq \frac{\tilde{\lambda}_n}{\lambda} > 0.$$

Since $\dfrac{\tilde{\lambda}_k}{\tilde{\lambda}_k + \lambda} \leq 1$ and $\tilde{\lambda}_n > 0$, the minimum eigenvalue is:

$$\lambda_{\min} = \frac{\tilde{\lambda}_k}{\tilde{\lambda}_k + \lambda}.$$

**Step 6: Computing the Condition Number**

The condition number is given by:

$$\kappa(M_k^{-1}\mathcal{G}) = \frac{\lambda_{\max}}{\lambda_{\min}} = \frac{\dfrac{\tilde{\lambda}_{k+1}}{\lambda}}{\dfrac{\tilde{\lambda}_k}{\tilde{\lambda}_k + \lambda}} = \frac{\tilde{\lambda}_{k+1}(\tilde{\lambda}_k + \lambda)}{\tilde{\lambda}_k \lambda}.$$

**Conclusion:**

Thus, the exact condition number of the preconditioned system $M_k^{-1}\mathcal{G}$ is:

$$\kappa(M_k^{-1}\mathcal{G}) = \frac{\tilde{\lambda}_{k+1}(\tilde{\lambda}_k + \lambda)}{\tilde{\lambda}_k \lambda}.$$

$\square$

# B ADDITIONAL MATERIAL FOR THE EXPERIMENTS

## B.1 ADDITIONAL EXPERIMENT: THE ALLEN-CAHN EQUATION

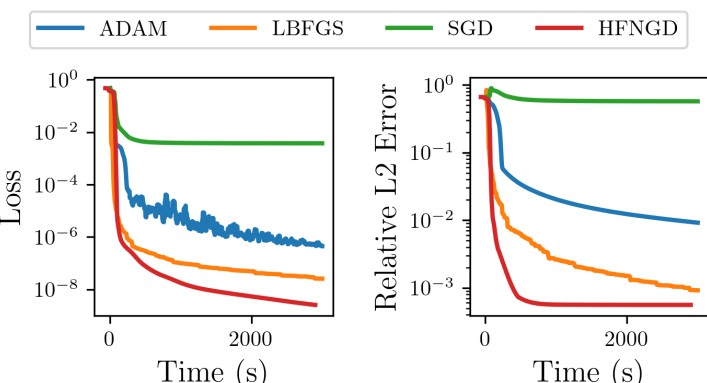

Figure 6: Loss and Relative L2 Error plotted against Time for different methods on the Allen-Cahn equation for the best run out of each optimizer. Our optimizer noticeably achieves the fastest convergence out of all optimizers.

We consider the Allen-Cahn equation, a representative case with which conventional PINN models are known to struggle. It takes the form:

$$u_t - 0.0001 u_{xx} + 5u^3 - 5u = 0, \quad t \in [0, 1],\ x \in [-1, 1],$$

$$u(0, x) = x^2 \cos(\pi x), \quad u(t, -1) = u(t, 1), \quad u_x(t, -1) = u_x(t, 1).$$

The network used is a modified MLP from Wang et al. (2023), with periodic embeddings and 4 hidden layers of 128 neurons each, resulting in approximately 51k trainable parameters.

While L-BFGS performs quite well on this task, achieving competitive convergence rates, our Hessian-Free Natural Gradient Descent (HF-NGD) method significantly outperforms it, particularly in terms of speed and final accuracy. As shown in Figure 6, HF-NGD achieves much faster convergence, and the final relative $L^2$ error is approximately five times lower than that of L-BFGS.

| Solver | Best | | Worst | | Median | |
|---|---|---|---|---|---|---|
| | L2 Error | Loss | L2 Error | Loss | L2 Error | Loss |
| ADAM | 9.1279e-03 | 4.3189e-07 | 9.3110e-03 | 1.1468e-06 | 9.2569e-03 | 6.0420e-07 |
| LBFGS | 9.1654e-04 | 2.5690e-08 | 9.1740e-04 | 2.5670e-08 | 9.1690e-04 | 2.5660e-08 |
| SGD | 5.7104e-01 | 3.7522e-03 | 5.7110e-01 | 3.7527e-03 | 5.7108e-01 | 3.7526e-03 |
| HF-NGD | 5.5788e-04 | 2.5640e-09 | 5.8536e-04 | 2.9216e-08 | 5.8518e-04 | 2.9092e-08 |

Table 2: Best, worst, and median final relative L2 error and loss for each solver in the Allen-Cahn experiment.

| Hyperparameter | Value |
|---|---|
| Network Architecture | Modified MLP with Periodic Embeddings |
| Embedding Period | 2.0 |
| Layer Sizes | 4 hidden layers (128 neurons each) |
| Trainable Parameters | 50,304 |
| Levenberg-Marquardt Damping | $\min(1 \times 10^{-5}, \text{loss})$ |
| Lanczos Steps ($N_L$) | 10 |
| Initial Rank for Adaptive Steps | 250 |
| Adaptive Rank Increment | 50 |
| Max Rank for Eigen Decomposition | 1500 |
| Grid Points (Interior) | 262,144 |
| Grid Points (Boundary) | 32,768 |
| CG Tolerance | $\min(1 \times 10^{-5}, \text{loss})$ |
| Time Budget | 3000 seconds |
| **Adam Optimizer** | Learning Rate: $1 \times 10^{-3}$
Warmup Steps: 1000 (Linear Scheduler)
Decay Steps: 20,000 (Exponential Decay Scheduler)
Decay Rate: 0.5 |
| **L-BFGS Optimizer** | Maximum Iterations: 50,000
Tolerance: $1 \times 10^{-5}$ |
| **SGD Optimizer** | Learning Rate: $5 \times 10^{-3}$
Momentum: 0.9
Gradient Clipping: 1.0 |

Table 3: Hyperparameters for the Allen-Cahn equation experiment

## B.2 RESIDUES AND LINEARIZED RESIDUES

| Equation | Residual, $R(u_\theta)$ | Linearized Residual, $DR(u_\theta)[\delta u_\theta]$ |
|:---:|:---|:---|
| **Allen-Cahn** | $u_t - d \cdot u_{xx} + 5(u^3 - u) - f$ | $\frac{\partial \delta u_\theta}{\partial t} - d \cdot \frac{\partial^2 \delta u_\theta}{\partial x^2} + (15u_\theta^2 - 5)\delta u_\theta$ |
| **2D Klein-Gordon** | $u_{tt} - u_{xx} - u_{yy} + u^2 - f$ | $\frac{\partial^2 \delta u_\theta}{\partial t^2} - \frac{\partial^2 \delta u_\theta}{\partial x^2} - \frac{\partial^2 \delta u_\theta}{\partial y^2} + 2u_\theta \delta u_\theta$ |
| **Poisson** | $-\nabla^2 u - f$ | $-\nabla^2 \delta u_\theta$ |
| **Navier-Stokes (Vorticity)** | $\omega_t + \mathbf{u} \cdot \nabla\omega - \nu\nabla^2\omega - f$ | $\frac{\partial \delta\omega_\theta}{\partial t} + \mathbf{u} \cdot \nabla\delta\omega_\theta + \delta\mathbf{u}_\theta \cdot \nabla\omega - \nu\nabla^2\delta\omega_\theta$ |
| **Navier-Stokes (Velocity)** | $\frac{\partial \mathbf{u}}{\partial t} + \mathbf{u} \cdot \nabla\mathbf{u} + \nabla p - \nu\nabla^2\mathbf{u} - \mathbf{f}$ | $\frac{\partial \delta\mathbf{u}_\theta}{\partial t} + \mathbf{u} \cdot \nabla\delta\mathbf{u}_\theta + \delta\mathbf{u}_\theta \cdot \nabla\mathbf{u} + \nabla\delta p_\theta - \nu\nabla^2\delta\mathbf{u}_\theta$ |

Table 4: Residuals $R(u_\theta)$ and their linearized residuals $DR(u_\theta)[\delta u_\theta]$ for considered benchmarks: Allen-Cahn, 2D Klein-Gordon, Poisson, and Navier-Stokes in both vorticity and velocity forms.

## B.3 HYPERPARAMETERS FOR THE NAVIER-STOKES 2D STEADY-STATE EXPERIMENT

| Hyperparameter | Value |
|:---|:---:|
| Layer Sizes | [2, 100, 100, 100, 100, 3] |
| Activation Function | $\tanh$ |
| Reynolds Number (Re) | 40 |
| Viscosity ($\nu$) | 1/Re |
| Levenberg-Marquardt Damping | $\min(1, \text{loss})$ |
| Lanczos Steps ($N_L$) | 10 |
| Initial Rank for Adaptive Steps | 250 |
| Adaptive Rank Increment | 50 |
| Max Rank for Eigen Decomposition | 1500 |
| Seed for Random Initialization | 0 |
| Grid Points (Interior) | 2601 |
| Grid Points (Boundary) | 400 |
| CG Tolerance | $\min(1, \text{loss})$ |

Table 5: Hyperparameters for the Navier-Stokes 2D steady-state experiment

## B.4 ADDITIONAL RESOURCES FOR THE KLEIN-GORDON EQUATION EXPERIMENT

| Solver | Best | | Worst | | Median | |
|:---:|:---:|:---:|:---:|:---:|:---:|:---:|
| | L2 Error | Loss | L2 Error | Loss | L2 Error | Loss |
| ADAM | 2.0676e-03 | 9.8366e-07 | 8.4513e-03 | 5.5133e-06 | 3.5232e-03 | 3.7752e-06 |
| LBFGS | 1.1752e-04 | 5.2457e-09 | 7.4118e-04 | 1.0437e-08 | 3.6707e-04 | 1.4234e-08 |
| HF-NGD | 3.8747e-05 | 4.5910e-10 | 2.5555e-04 | 7.8761e-10 | 1.3485e-04 | 4.0539e-10 |
| SGD | 1.0860e-03 | 3.0317e-06 | 4.4600e-02 | 3.7092e-03 | 3.0289e-02 | 7.6686e-03 |

Table 6: Best, worst, and median final relative L2 error and loss for each solver in the Klein-Gordon experiment.

| Hyperparameter | Value |
|---|---|
| Network Architecture | Separable PINNCho et al. (2023) |
| Layer Sizes | 4 hidden layers (128 neurons per spatial dimension) |
| Trainable Parameters | 149,376 |
| Levenberg-Marquardt Damping | $\min(1 \times 10^{-5}, \text{loss})$ |
| Lanczos Steps ($N_L$) | 10 |
| Initial Rank for Adaptive Steps | 250 |
| Adaptive Rank Increment | 50 |
| Max Rank for Eigen Decomposition | 1500 |
| Grid Points (Interior) | 262,144 |
| Grid Points (Boundary) | 32,768 |
| CG Tolerance | $\min(1 \times 10^{-5}, \text{loss})$ |
| Time Budget | 6000 seconds |
| **HF-NGD Optimizer** | Learning Rate: $1 \times 10^{-3}$ 
 Warmup Steps: 1000 (Linear Scheduler) 
 Decay Steps: 20,000 (Exponential Decay Scheduler) 
 Decay Rate: 0.5 |
| **HFLR Optimizer** | Learning Rate: $5 \times 10^{-4}$ 
 Maximum Iterations: 50,000 
 Tolerance: $1 \times 10^{-5}$ |
| **Adam Optimizer** | Learning Rate: $1 \times 10^{-3}$ 
 Warmup Steps: 1000 (Linear Scheduler) 
 Decay Steps: 20,000 (Exponential Decay Scheduler) 
 Decay Rate: 0.5 |
| **L-BFGS Optimizer** | Maximum Iterations: 50,000 
 Tolerance: $1 \times 10^{-5}$ |
| **SGD Optimizer** | Learning Rate: $5 \times 10^{-3}$ 
 Momentum: 0.9 
 Gradient Clipping: 1.0 |

Table 7: Hyperparameters for the Klein-Gordon equation experiment and associated solvers

## B.5 ADDITIONAL RESOURCES FOR THE POISSON EQUATION IN 10 DIMENSIONS

| Solver | Best | | Worst | | Median | |
|---|---|---|---|---|---|---|
| | L2 Error | Loss | L2 Error | Loss | L2 Error | Loss |
| ADAM | 4.2226e-03 | 4.4942e-06 | 1.0286e-02 | 9.9630e-05 | 5.4410e-03 | 3.1723e-05 |
| LBFGS | 3.2373e-04 | 2.6123e-08 | 4.8860e-04 | 3.0253e-08 | 4.7009e-04 | 4.4788e-08 |
| HF-NGD | 3.2020e-05 | 4.0725e-10 | 7.1201e-05 | 1.9829e-09 | 3.8316e-05 | 3.4561e-10 |
| SGD | 5.0698e-03 | 1.1918e-05 | 3.2487e-02 | 1.5308e-03 | 6.9154e-03 | 3.6156e-05 |

Table 8: Best, worst, and median final relative L2 error and loss for each solver in the Poisson equation in 10 dimensions experiment.

| Hyperparameter | Value |
|---|---|
| Network Architecture | MLP (5 hidden layers, Tanh activations) |
| Layer Sizes | [10, 512, 512, 512, 512, 512, 1] |
| Trainable Parameters | 1,102,849 |
| Lanczos Steps ($N_L$) | 10 |
| Initial Rank for Adaptive Steps | 250 |
| Adaptive Rank Increment | 150 |
| Max Rank for Eigen Decomposition | 1500 |
| Levenberg-Marquardt Damping | $\min(1 \times 10^{-5}, \text{loss})$ |
| Grid Points (Interior) | 262,144 |
| Grid Points (Boundary) | 32,768 |
| Integrator Resampling Frequency | Every 250 epochs |
| Interior Integrator Sample Size | 4000 |
| Boundary Integrator Sample Size | 500 |
| Evaluation Integrator Sample Size | 4000 |
| Time Budget | 3000 seconds |
| **Adam Optimizer** | Learning Rate: $1 \times 10^{-3}$
Warmup Steps: 1000 (Linear Scheduler)
Decay Steps: 20,000 (Exponential Decay)
Decay Rate: 0.5 |
| **L-BFGS Optimizer** | Maximum Iterations: 50,000
Tolerance: $1 \times 10^{-5}$ |
| **SGD Optimizer** | Learning Rate: $5 \times 10^{-3}$
Momentum: 0.9 |

Table 9: Hyperparameters for the Poisson equation in 10 dimensions experiment

## B.6    ADDITIONAL RESOURCES FOR THE VORTICITY EQUATION EXPERIMENT

| Solver | Best | | Worst | | Median | |
|---|---|---|---|---|---|---|
| | L2 Error | Loss | L2 Error | Loss | L2 Error | Loss |
| ADAM | 9.0281e-03 | 1.4407e-04 | 9.0281e-03 | 1.4407e-04 | 9.0281e-03 | 1.4407e-04 |
| LBFGS | 1.8595e-03 | 1.1438e-06 | 5.4369e-02 | 1.4413e-03 | 1.8636e-03 | 1.1486e-06 |
| HF-NGD | 9.0018e-05 | 1.1977e-08 | 9.0810e-05 | 1.2204e-08 | 9.0137e-05 | 1.2011e-08 |
| SGD | 1.8706e-02 | 4.4375e-04 | 1.8706e-02 | 4.4375e-04 | 1.8706e-02 | 4.4375e-04 |

Table 10: Best, worst, and median final relative L2 error and loss for each solver in the Kovaznay experiment.

| Hyperparameter | Value |
|---|---|
| Branch Network Architecture | MLP (6 hidden layers, Tanh activations) |
| Branch Network Layer Sizes | [1, 100, 100, 100, 100, 100, 100] |
| Trunk Network Architecture | MLP (6 hidden layers, Tanh activations) |
| Trunk Network Layer Sizes | [2, 100, 100, 100, 100, 100, 100] |
| Total Trainable Parameters | 101,500 |
| Batch Size | 256 |
| Reynolds Number Range | 50 to 250 |
| Reynolds Number Validation | 500 |
| Training Time Budget | 3000 seconds |
| Grid Points (Interior) | 2601 |
| Grid Points (Boundary) | 400 |
| Lanczos Steps (HF-NGD) | 10 |
| Initial Rank for Adaptive HF-NGD | 250 |
| Adaptive Rank Increment | 50 |
| Max Rank for Eigen Decomposition | 1500 |
| Levenberg-Marquardt Damping (HF-NGD) | $\min(1 \times 10^{-5}, \text{loss})$ |
| Learning Rate (Adam) | $1 \times 10^{-3}$ |
| Learning Rate (SGD) | $1 \times 10^{-2}$ |
| Learning Rate Decay (Adam/SGD) | Exponential (Decay Rate: 0.96 every 1000 steps) |
| SGD Momentum | 0.9 |
| Gradient Clipping Threshold (SGD) | 1.0 |
| Optimizer Tolerance (BFGS) | $1 \times 10^{-5}$ |

Table 11: Hyperparameters for the DeepONet applied to the Vorticity Equation in Kovasznay flow configuration.

