# OpenReview forum: "Hessian-Free Natural Gradient Descent for Physics Informed Machine Learning"
_ICLR.cc/2025/Conference — ICLR 2025 Conference Withdrawn Submission_

### Official Review · Reviewer_7CZX · 2024-10-20

**Soundness:** 3
**Presentation:** 3
**Contribution:** 3
**Rating:** 6
**Confidence:** 3

**Summary:**

This paper introduced a Hessian-Free Natural Gradient Descent (HF-NGD) framework for Physics-Informed Machine Learning including PINNs and neural operators. The method is matrix-free, with additional memory cost for iterations. The method achieves state-of-the-art performance on different benchmarks, 1~2 orders improvement compared to SGD and LBFGS , and over a 2-orders improvement on neural operators. This paper is the first to apply low-rank preconditioning in Gauss-Newton methods for PINN and neural operator optimization, making contributions to the community of physics-informed machine learning.

**Strengths:**

1. The originality comes from the combination of preconditioning technique, the Matrix-free Hessian approximation, PCG and line search, making an explicit experimental improvement on the prediction accuracy. The comparison to UGNNG and DGNNG shows the importance of preconditioning.
2. The paper is well-written and easy to follow, both theoretical and experimental richly.
3. The paper makes contribution to the second-order optimization techniques on physics-informed machine learning.

**Weaknesses:**

1.The contributions should be clarified more clearly compared to existing second order optimization techniques. The title is “Hessian free …” but there are works without explicitly forming the Hessian matrix, and the improvements of this paper may mainly comes from the preconditioning?

2.Since the method is more like a combination of mathematical techniques, the ablation study should be reported, to show the independent importance of preconditioning, matrix-free computation cost, and so on.

3.Section 3.2 and 3.3 are mathematically expressed independently, and the relation to Hessian matrix and physics-informed machine learning should be clarified more clearly to make the paper more readable.

**Questions:**

see in the weaknesses.

---

### Official Review · Reviewer_ynu6 · 2024-11-01

**Soundness:** 2
**Presentation:** 1
**Contribution:** 2
**Rating:** 3
**Confidence:** 2

**Summary:**

This paper introduces a Hessian-Free Natural Gradient Descent (HF-NGD) framework tailored for Physics-Informed Machine Learning (PIML), aiming to overcome optimization difficulties in solving partial differential equations (PDEs) with Physics-Informed Neural Networks (PINNs). By circumventing the need to compute and store full Hessian matrices, the approach achieves improved efficiency and scalability, even for larger neural networks and complex PDEs. It also incorporates a preconditioning strategy that significantly enhances convergence rates. The proposed method is evaluated on several benchmark PDE problems, demonstrating substantial improvements in both speed and accuracy over standard optimizers like LBFGS and Adam.

**Strengths:**

1. The HF-NGD framework’s use of a matrix-free Gauss-Newton approximation provides a highly efficient and scalable solution for training PINNs, overcoming traditional computational constraints associated with second-order methods.

2. The preconditioning approach leverages spectral properties of the Gauss-Newton Hessian, efficiently addressing common issues like ill-conditioning and facilitating faster convergence in Conjugate Gradient (CG) solvers.

**Weaknesses:**

1. While the HF-NGD method is tested on a 10-dimensional Poisson equation, it would be valuable to see experiments on even higher-dimensional PDEs. Since many real-world machine learning problems are inherently high-dimensional, demonstrating the scalability of the approach in such settings would significantly bolster the method’s impact and relevance.

2. The inconsistent citation formatting, especially the omission of parentheses where needed, disrupts the flow of the paper and makes it difficult to follow. Correcting these inconsistencies would improve readability.

3. There are several typographical and formatting errors in the paper, such as “Eq. equation,” “THE GAUSS-NEWTON GAUSS-NEWTON HESSIAN,” “section4.1,” and “minimaland”. Careful proofreading would help ensure a polished and professional presentation.

4. The authors mention that their proposed method has a faster convergence rate. What is that rate?

5. What is Lemma 3.2, whose proof is shown in an appendix? Does the implication of step 4 always hold true: $M_k v_i = m_i v_i \implies M_k^{-1} v_i = \frac{1}{m_i} v_i$ ? What happens if $M_k$ is not invertible and m=0?

6. Why did the authors choose those specific PDE problems to show the efficiency of their method? How will the proposed method work in the case of Burgers, Advection, Euler-Bernoulli, and nonlinear diffusion, to mention a few?

7. Do the plots on all Figures show the results for ten random seeds or just one? Plotting the mean and standard deviation of different random seeds would be best.

8. How did the authors use the Sherman-Morrison-Woodbury formula to have $M^{-1}_k = \frac{1}{\lambda} I - \frac{1}{\lambda^2} V_k D^{-1} V_k^\top$?  Why in this equation: $M^{-1}_k v = \frac{1}{\lambda} v - \frac{1}{\lambda^2} V_k D V_k^\top v$ is used $ D$ and not $D^{-1} $? Also, is there a guarantee that D will always be invertible?

**Questions:**

.

---

### Official Review · Reviewer_z9WC · 2024-11-01

**Soundness:** 3
**Presentation:** 4
**Contribution:** 3
**Rating:** 5
**Confidence:** 4

**Summary:**

This paper investigates a Hessian-free natural gradient descent (NGD) method designed for efficient training of PINNs. Instead of computing the matrix inverse directly, the proposed algorithm first applies preconditioning to the linear system by low-rank decomposition, followed by solving the system with matrix-vector products. Theoretical and empirical results support the effectiveness of preconditioning in improving the conditioning number of the linear system.

**Strengths:**

(1) The method is reasonable and well-founded, as the low-rank structure can reduce the computational costs, the preconditioning can improve the condition number, the matrix inverse can be approximately calculated by matrix-vector products. Although all those techniques are quite common in computational mathematics, heir combination in the context of Hessian-free NGD is interesting and potentially impactful.

(2) Empirical results show that the proposed method can significantly reduce training loss magnitude, demonstrating its potential effectiveness.

**Weaknesses:**

It seems that the proposed algorithm is not designed purely for PINNs. It does not make use of the specific properties and structures of PINNs, suggesting it may be applicable to broader machine learning tasks. All results are not specifically for PINNs. This raises a question regarding the motivation for developing HF-NGD specifically for PINNs.

**Questions:**

Have there been previous works investigating Hessian-free NGD for general regression tasks? Are you applying existing algorithms specifically for PINNs, or is the main contribution here the empirical application of HF-NGD to PINNs? While most references provided in the introduction focus on PINNs, I would be interested in understanding the development and current results of HF-NGD more broadly. It would be helpful to clarify if similar algorithms already exist and whether your primary contribution lies in adapting them for PINNs.

If the authors can clearly articulate their main contributions and provide a more detailed comparison with existing methods, I would be happy to raise my score as I really appreciate using the techniques in applied math  for machine learning. My primary concern is whether similar algorithms already exist for general tasks, and this paper is primarily applying those methods to PINNs.

---

### Official Review · Reviewer_xLjj · 2024-11-03

**Soundness:** 1
**Presentation:** 3
**Contribution:** 2
**Rating:** 3
**Confidence:** 4

**Summary:**

This work proposes a second-order optimization algorithm for PINNs. They do so by considering matrix-free algorithms for approximating the Gauss-Newton updates. The main challenge in this approach is that the Gauss-Newton matrix is highly ill-conditioned for Neural Networks. They attempt to alleviate this problem using low-rank approximation of the Gauss-Newton as preconditioners for the Linear System. They perform experiments on various PDEs such as 2D Navier Stokes, Poisson Equation etc benchmarking against ADAM, LBFGS and SGD.

**Strengths:**

1. The idea of using matrix-free algorithms is a very good one and should be explored more thoroughly. This method should be scalable to large-scale problems because it avoids the need to compute or store the full Hessian matrix. This work also correctly identifies that the main difficulty in this approach is finding good preconditioners for the algorithm.

2. The paper is clearly written and well structured. Empirical evaluations are presented well and experiments include neural networks with up to a million trainable parameters.

3. Despite the problems with the method section outlined below, I still find results impressive, specially the significantly faster convergence in figures 3 and 4.

**Weaknesses:**

## Novelty

The idea of using matrix-free algorithms to approximate second-order updates is not exactly groundbreaking. As mentioned by the authors as well, it is well-known in pre-existing works. However, the authors also propose a preconditioning strategy for these algorithms, which I believe is a novel and interesting contribution, since it is non-trivial to find good preconditioners for these systems. This is why my review focuses on the correctness of the proposed preconditioner

## Correctness

1. Authors show a plot of the spectrum of a small MLP in Figure 1 and claim that it is representative of the spectrum of Gauss-Newton matrices in general. There have been previous works analyzing the spectrum of Gauss-Newton/Fisher matrices in a lot more details such as [1], which show that taking care of the top few eigenvalues is not sufficient. Even considering the analysis in the paper, from figure 1 it should be clear that taking care of some of the leading eigenvalues will still give us an ill-conditioned system. While iit's true that Gauss-Newton is generally low-rank, it is not *that* low rank. Concretely it can be written as a sum of N rank O matrices, where N is size of the dataset and O is the output dimensions. Hence the rank of the Gauss-Newton is upper bounded by NO which can be quite big. So it is unlikely that for large problems it will be possible to take care of all the necessary leading eigenvalues.

2. It is unclear what is the justification for alternating between orthogonal iteration and lanczos decomposition. I don't even see the computational benefits of this because computing residual norm in the orthogonal iteration requires matrix-matrix multiplication between $\mathcal{H}$ and the previous lanczos vectors which is the same number of matvecs as required for Lanczos Decomposition. Lanczos Decomposition will have additional cost of reorthogonalization but for large neural networks the cost of matvecs should dominate.

3. My biggest concern with this paper is Lemma 3.2 about the conditioning number of the preconditioned system. This seems like a key result in this work, since the preconditioning strategy is based on this result. As stated the lemma is simply false. Moreover, the proof of this lemma in the appendix is full of very strange errors. Firstly I should say deriving the conditioning number of the preconditioned system is trivial. Since $M_k$ and $\mathcal{H}$ can be said to have the same eigenvectors, the eigenvalues of $M_k^{-1}\mathcal{G}$ is given by $(\frac{\lambda_1}{\lambda_1 + \lambda}, \dots, \frac{\lambda_k}{\lambda_k + \lambda}, \frac{\lambda_{k+1}}{\lambda}, \dots, \frac{\lambda_n}{\lambda} )$. The largest eigenvalue is given by $\lambda_{max} = max( \frac{\lambda_{k+1}}{\lambda}, \frac{\lambda_1}{\lambda_1 + \lambda})$ and $\lambda_{min} = min( \frac{\lambda_{n}}{\lambda}, \frac{\lambda_k}{\lambda_k + \lambda})$. The proof presented in the appendix presents this one line result in 2 pages making several errors in the process. In Step 2 the decomposition of $M_k$ in the first equation completely makes no sense. Somehow after "Simplifying" they end with the correct decomposition even though the previous equation clearly doesn't simplify to this. Steps 3 and 4 are ok even if unnecessarily verbose. Step 5 is again completely wrong where despite writing down the spectrum they are unable to correctly identify the maximum and minimum eigenvalues which I find very strange. A lot of the reasoning here is nonsensical and irrelevant.

## Experimental details
In appendix B.3 it is stated that the CG tolerance is chosen to be 1. This seems a bit high to me.

## Minor Factual errors:
In 161 it is stated "This approach circumvents the prohibitive cubic cost ...". This is false because in CG matvecs cos O(n^2) and requiring O(n) iterations for convergence. Hence it is still cubic. The main advantage is the lower space complexity.



[1] Papyan, Vardan. "Traces of class/cross-class structure pervade deep learning spectra." Journal of Machine Learning Research 21.252 (2020): 1-64.

**Questions:**

See weaknesses

---

### Official Review · Reviewer_zQeC · 2024-11-04

**Soundness:** 4
**Presentation:** 4
**Contribution:** 4
**Rating:** 8
**Confidence:** 4

**Summary:**

This paper presents a computational method to reduce the computational overhead of the Energy Natural Gradient method introduced by Müller et al. (see https://proceedings.mlr.press/v202/muller23b.html), as well as the Gauss-Newton Natural gradient introduced in Jnini et al. (see https://arxiv.org/abs/2402.10680) and further refined in Müller et al. (see https://icml.cc/virtual/2024/poster/34242).
Both Natural Gradient and Gauss-Newton Natural gradient rely on the estimation and inversion of a dense square matrix $\mathcal{H}(\theta)$ whose dimension corresponds to the number of neural network parameters, which makes the method prohibitive for large neural networks, a well-known problem of natural gradient methods in general.
The authors propose to circumvent this difficulty by using a conjugate gradient method. More formally, given the Jacobian $J(\theta)$ of the total empirical residual defined in section 2.2, we have the estimate $\mathcal{H}(\theta) = J(\theta)^T J(\theta)$.
We can thus replace the inversion of $\mathcal{H}(\theta)$ in order to calculate $\mathcal{H}(\theta)^+\nabla L(\theta)$ by the successive resolution of $J(\theta)^Tw=\nabla L(\theta)$ then $J(\theta)v=w$ by the conjugate gradient method.
Unfortunately, the conjugate gradient method struggles to converge when the matrix is ill-conditioned. The authors therefore propose a preconditioning of the method, based on a low-rank approximation of the inverse of the matrix $\mathcal{H}(\theta)$, computed through an adaptation of the Lanczos decomposition and the orthogonal decomposition, which effectively control the rank of the approximation, coupled with the Sherman-Morrison-Woodbury inversion formula.
The result is a method that scales reasonably well in relation to the number of parameters, as shown by the various empirical results.

**Strengths:**

- Clear and concise writing
- The presented method successfully addresses the scalability limitation of previous natural gradient methods for pinns, which until now prevented their application to more complex problems.
- The paper presents a simple and effective mechanism for adapting order of the low rank approximation, by adapting classical numerical linear algebra algorithms.

**Weaknesses:**

- In section 2.2, the discretization process may be exposed more explicitly. As it currently stands, the empirical residue is not explicitly shown, which can make it difficult to understand.
- In section 2.3 first equation : I think there is a diagonal scaling matrix missing between $J(\theta)^T$ and $J(\theta)$. (with first entries being equal to $\frac{1}{N_\Omega}$ and last entries equals to $\frac{1}{N_{\partial\Omega}}$, with $N_\Omega$ being the number of points in the domain and $N_{\partial\Omega}$ the number of points on the boundary).
- In section 2.3 again, there is prbably a typo : $\mathcal{H}v=J^Tw$ should be replaced by $\mathcal{H}\nabla L(\theta)=J^Tw$

**Questions:**

- I find the term “Hessian-free” a little confusing, as you're still using a low-rank approximation of the Hessian. In particular, it would be interesting to compare your approach with the use of the low-rank approximation of the Hessian directly as the pseudo-inverse of $\mathcal{H}(\theta)$, since this would further reduce the cost of your algorithm.

---

### Note · Authors · 2024-11-17

I have read and agree with the venue's withdrawal policy on behalf of myself and my co-authors.